# Ambrosia Beetles Prefer Closed Canopies: A Case Study in Oak Forests in Central Europe

Jaroslav Holuša [1] , Tomáš Fiala [1,]*  and Jiří Foit [2]

1   Faculty of Forestry and Wood Sciences, Czech University of Life Sciences, Kamýcká 129, 16500 Praha, Czech Republic; holusaj@seznam.cz
2   Faculty of Forestry and Wood Technology, Mendel University in Brno, Zemědělská 3, 61300 Brno, Czech Republic; foit.jiri@gmail.com
*   Correspondence: tomas.fiala@nature.cz; Tel.: +42-07-2415-1113

**Abstract:** Research Highlights: The percentage of canopy closure was found to be the main factor associated with ambrosia beetle abundance and species richness. The latter two variables increased as canopy closure increased, probably because a high percentage of canopy closure provides a stable and humid environment suitable for the growth of ambrosia fungi. Objectives: Oak is a common host tree for ambrosia beetles (Coleoptera: Curculionidae: Scolytinae), which have independently evolved a nutritional mutualism with fungi. We suspected that ambrosia beetles might have specific habitat preferences that are different from those of other saproxylic beetles and that reflect the specific habitat preferences of their food, i.e., ambrosia fungi. Methods: We assessed ambrosia beetle abundance with ethanol-lured traps in five old-growth oak dominated forests and five managed oak dominated forests (one trap per forest) during the vegetation period in 2020. We determined whether ambrosia beetle abundance and species richness depend on forest type (managed vs. unmanaged), degree of canopy closure, abundance of oak trees, abundance of coarse deadwood, and abundance of dead oak branches. Results: In total, 4137 individuals of six species of ambrosia beetles associated with oaks were captured. The native ambrosia beetle *Anisandrus dispar* represented the majority of trapped ambrosia bark beetles. *A. dispar* along with another ambrosia beetle, *Xyleborinus saxesenii*, represented 99% of all captured beetles. Conclusions: In addition to canopy closure, the abundance of oak trees and the abundance of dead oak branches were significantly associated with ambrosia beetle abundance and species richness. The abundance of *A. dispar* was mainly correlated with dead oak branch abundance and the degree of canopy closure, whereas the abundances of *X. saxesenii* and of the invasive species *Xyleborinus attenuatus* and *Cyclorhipidion bodoanum* were mainly correlated with the net area occupied by oak trees.

**Keywords:** *Anisandrus dispar*; *Cyclorhipidion bodoanum*; deadwood; invasive species; *Xyleborus saxesenii*; *Xyleborinus attenuatus*; *Xylosandrus germanus*; Scolytinae; Quercus





## 1. Introduction

European temperate oak woodlands have a rich and unique biodiversity, which can be mainly attributed to the life history traits and structural characteristics of the oak trees *Quercus robur* and *Q. petraea* [1–3]. In Central Europe, oak trees are components of temperate broadleaf and mixed forests. Oak is a common host tree for ambrosia beetles (Coleoptera: Curculionidae: Scolytinae) [4,5].

The ambrosia beetles have independently evolved a nutritional mutualism with fungi [6]. Most species of ambrosia beetles depend on recently dead or stressed woody plants in which the beetles bore their tunnel systems ("galleries"). In the galleries, ambrosia beetles actively farm one or several fungal mutualists, which serve as their essential food source [6]. Some species of ambrosia beetles are among the most damaging forest pests, and species of quarantine significance are frequently moved intercontinentally [7,8].

Because oak trees attacked by other bark beetles provide suitable breeding substrates and other resources for ambrosia beetles, the volume of oaks and the area occupied by oaks under attack by other bark beetles are likely to affect the abundance of ambrosia beetles in a region [9]. Unlike other bark beetles that can fly tens of kilometers [10], ambrosia beetles can fly only up to about a hundred meters to a few kilometers [11,12]. The short dispersal distance of ambrosia beetles also suggests that the abundance of ambrosia beetles in an oak forest should be affected by the area that is occupied by oak.

Many studies in boreal and temperate forests have indicated that insect diversity increases as stands become more open because of higher temperatures and other changes in the microclimate [13–16]. Sun exposure was found to be the most important factor affecting the composition of buprestids and cerambycids in oak forests [17]. There has been sparse research on the influence of the canopy closure on the occurrence of scolytids, but some research suggests that the effects of canopy closure may differ among scolytid species. For example, *Scolytus intricatus* Ratzeburg, 1837 prefers oaks with a high canopy closure [18], but *Scolytus mali* Bechstein, 1805 prefers orchards with open canopies [19]. Similarly, the position of trap in the forest (edge vs. interior) also generally does not affect the occurrence of bark beetles [20–22], but it does affect the occurrence of some species. The scolytids *Hylurgops palliatus* Gyllenhal, 1813 and *H. glabratus* Zetterstedt, 1828, for example, require the shaded environment of the forest interior for feeding and are found in higher numbers in the forest interior than at the forest edge [23]. Two ambrosia beetles showed opposite patterns patterns: *Xylosandrus crassiusculus* Motschulsky, 1866 is more common at the forest edge, while *Xyleborinus andrewesi* Blandford, 1896 is more common in the forest interior [24]. Similarly, the bark beetle *Hylesinus taranio* Danthoine, 1788 prefers canopy closure at the forest edge [25]. We do not know of research focusing on the effect of canopy closure on ambrosia beetles.

Based on our unpublished observations, however, we suspect that ambrosia beetles are not primarily dependent on the sun exposure provided by an open canopy. On the contrary, as wet and warm conditions are important for the growth of their symbiotic fungi [26], ambrosia beetles are likely to be more abundant in wetter and warmer localities [27,28]. Localities are likely to be wetter and to have a more stable microclimate if the canopy is substantially closed rather than open [29].

The biodiversity of phloeoxylophagous insects is greater in old-growth oak stands than in many other kinds of forest stands, because old-growth oak stands have more deadwood, including dry branches in treetops [30,31]. For ambrosia species, this dependence was confirmed only in beech stands in that the abundance of ambrosia beetles was found to be higher in unmanaged than in managed stands [32].

In the current study, we tested the hypotheses that ambrosia beetle occurrence will depend on the degree of canopy closure, the abundance of oak trees, the abundance of deadwood, and abundance of dead oak branches. We also tested the hypothesis that ambrosia beetle occurrence is greater in unmanaged oak forests than in commercial forests [31,33,34].

## 2. Materials and Methods

### 2.1. Study Plots

The study was conducted in the biogeographically isolated area (the Chebsko-sokolovský bioregion) of "Western European broadleaf forests" in the western Czech Republic (Figure 1) (https://en.wikipedia.org/wiki/Western_European_broadleaf_forests, accessed on 12 August 2021). At lower altitudes in this area, "Western European broadleaf forest" is the main forest type. The natural vegetation of the bioregion consists mainly of acidophilic oaks (*Quercion roboris* Malcuit, 1929), but only 6% of the region is currently occupied by oaks. These are mostly in commercial oak forests, and old-growth oak forests have survived in only a few localities [35].

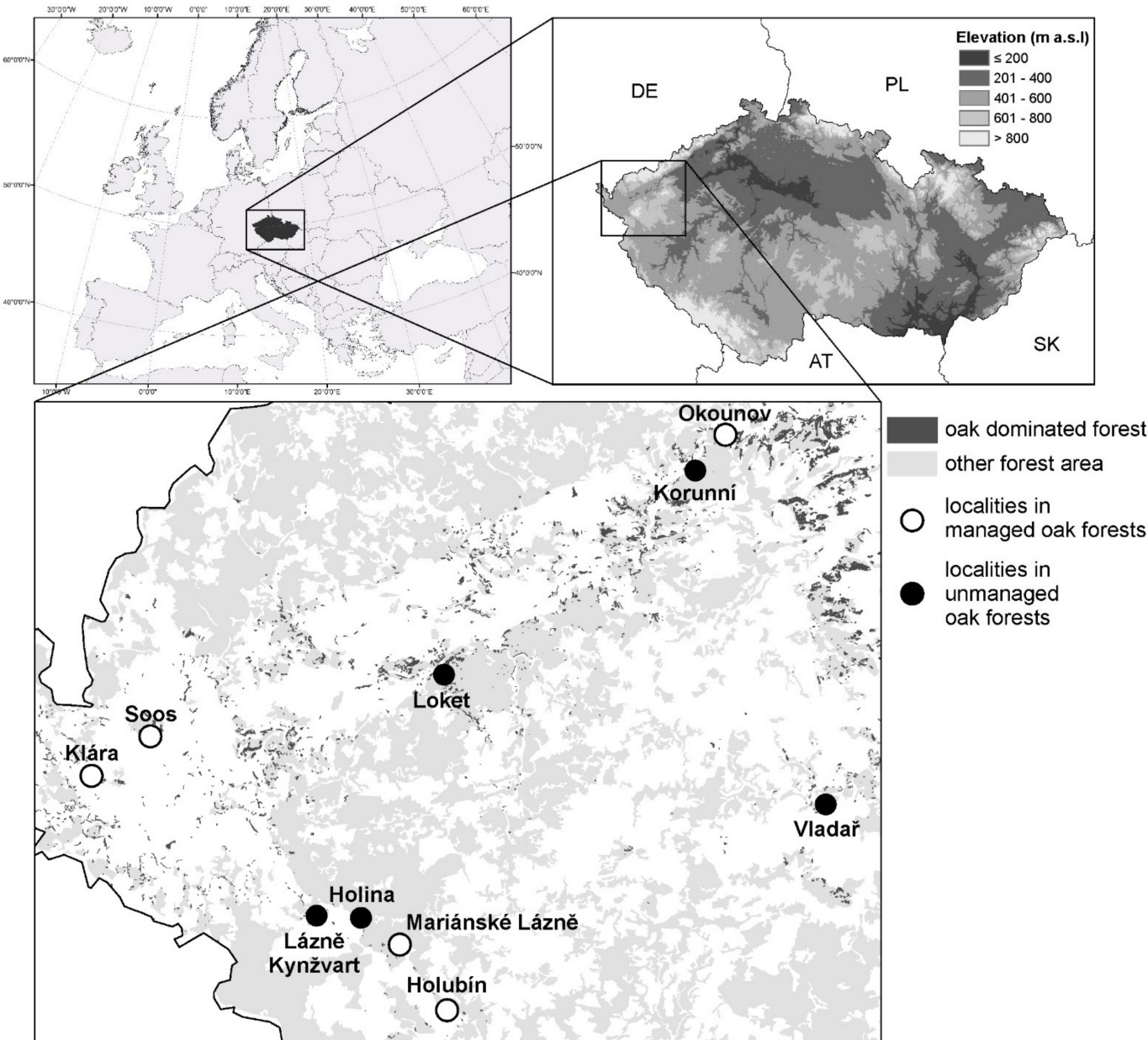

**Figure 1.** Locations of unmanaged oak dominated forests and managed oak dominated forests in western Bohemia where ambrosia beetle abundance and species richness were determined.

For assessing ambrosia beetle abundance by trapping, we selected oak dominated forests in which oaks represented > 60% of the trees (Only in study plot Soos the oak representation is 40%, and the rest of forest is cover by birch *Betullus* sp.) and that were >80 years old and ≥1 ha. The forests were at altitudes between 400–700 m a.s.l. (Table 1, Figure 1). The study plot is located in the mesophytic zone, which is characterized by an average annual rainfall of around 550–700 mm and an average annual temperature of 7.5 °C. Orographically, the study plots are located on flat land, with only the Korunní and Loket plots (northwest orientation) and the Vladař plot (south orientation) being on a steep slope.

**Table 1.** Characteristics of the studied oak dominated forests.

| Study Plot | Longitude; Latitude | Altitude (m a.s.l.) | Age | Forest Type | DBH (cm) | Net Area Occupied by Oaks (ha) | Volume of Oak Wood (m³/ha) | Volume of of Coarse Deadwood (m³/10 m²) | Abundance of Dead Oak Branches (No. per Tree) | Canopy Closure (%) | Distance from Stand Boundary (m) |
|---|---|---|---|---|---|---|---|---|---|---|---|
| Vladař | 50°4′31″ N, 13°12′33″ E | 605 | 130 | Unmanaged | 40 | 5.2 | 264 | 30 | 3 | 66 | 100 |
| Mariánské Lázně | 49°58′50″ N, 12°41′40″ E | 700 | 185 | Managed | 58 | 0.5 | 369 | 0 | 0.1 | 48 | 20 |
| Soos | 50°8′51″ N, 12°24′19″ E | 440 | 80 | Managed | 30 | 0.3 | 131 | 20 | 0.2 | 31 | 20 |
| Okounov | 50°21′45″ N, 13°6′28″ E | 440 | 80 | Managed | 40 | 0.5 | 58 | 5 | 0 | 35 | 10 |
| Korunní | 50°20′9″ N, 13°4′11″ E | 500 | 150 | Unmanaged | 50 | 1.5 | 262 | 20 | 1 | 66 | 50 |
| Klára | 50°7′7″ N, 12°19′59″ E | 440 | 90 | Managed | 40 | 0.3 | 215 | 20 | 0.2 | 45 | 20 |
| Holubín | 49°55′44″ N, 12°44′53″ E | 615 | 90 | Managed | 35 | 1.8 | 255 | 0 | 0.33 | 63 | 20 |
| Loket | 50°11′13″ N, 12°45′33″ E | 410 | 110 | Unmanaged | 20 | 2.5 | 123 | 15 | 1 | 64 | 40 |
| Holina | 50°0′8″ N, 12°38′58″ E | 700 | 180 | Unmanaged | 50 | 0.5 | 361 | 50 | 5 | 68 | 30 |
| Lázně Kynžvart | 50°0′19″ N, 12°35′48″ E | 565 | 200 | Unmanaged | 120 | 0.7 | 245 | 5 | 1 | 66 | 10 |

*Quercus robur* was dominant in all localities except Vladař and Loket (see Figure 2), where both *Q. petraea* and *Q. robur* grow but *Q. petraea* dominates.

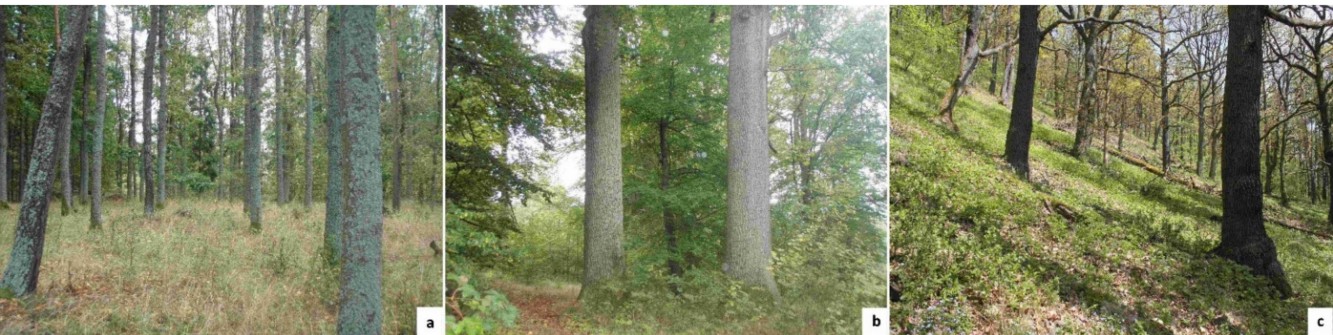

**Figure 2.** Views of managed 80-year-old oak dominated forests at Holubín (**a**) and unmanaged oak forests at Kynžvart (**b**) and Vladař (**c**).

### 2.2. Traps and Lures

To estimate the abundance and species richness of ambrosia beetles [4,36], one trap baited with ethanol was placed in the center of each study plot. Ethanol was released from a plastic-vial dispenser (ca. 250 mg·day$^{-1}$). These dispensers were made of polyethylene with foam and were 5 cm in diameter and 5 cm high. Each dispenser was placed in a Theysohn® trap that was located ca. 1.3 m above the ground and that faced the main wind direction. The traps were emptied, and the ethanol was replaced every 2 weeks from the beginning of April to the end of August in 2020. All trapped insects were preserved in 70% ethanol.

The insects were identified by the second author, who used Pfeffer's key [37]. Dr. Miloš Knížek (Prague) confirmed the identification of *C. bodoanum*.

### 2.3. Environmental Variables

Forest type. We recognized two types of forests according to management. Mature-managed forests were oak dominated forests between 80 and 120 years old. All trees in each managed forest were the same age and were very homogenous; cut stumps were abundant. Forests in this category had reached maturity (i.e., had attained their maximum annual increase in volume) and represented typical state-owned forests. The volume of

deadwood and dead branches was low (Table 1). These forests were last managed ten years ago.

Most of the oldest trees in unmanaged forests were >120 years old, and the forests had not been managed for the last 70 years. The forests in this category represented the closest-to-natural forests that remain in western Bohemia. The only signs of human interference were a few scattered stumps from past selective cuttings. Unlike the managed forests, the unmanaged forests included trees of all ages including small areas with young trees. The volume of deadwood and dead branches was high (Table 1). These forests were last managed more than fifty years ago. The unmanaged forest at Kynžvart (Figure 2b) had been modified into a park with grasslands, but more than 60% of the area was covered with trees, which grew in large unbroken patches. Because the structure of this stand was otherwise similar to old-growth stands, we included this stand in the unmanaged forest category (Figure 2b). In contrast, the forest at Soos, although located in a protected area, was classified as a managed forest because it was a homogeneous stand that had been planted in a meadow.

Net area occupied by oaks: The net area was calculated as the total area of the stand multiplied by the tree density and the percentage represented by oak. Data were obtained from the regional inventory of forests.

Volume of oak wood: Data for the volume of oak trees ($m^3$/ha) in the oak dominated forests were obtained from the forest management plan, which contained detailed data for all forest stands.

Volume of coarse deadwood: Deadwood volume was quantified in five areas of $10\ m^2$/area. The diameters and lengths of the dead trees and dead branches were measured manually.

Abundance of dead oak branches: Numbers of large dead branches were determined on 10 oak trees along a transect running through each study area; the values were subsequently expressed as the mean number of dead branches per tree. The transects were located in the central part of each study area (one transect per study area) and were about 50-m long. Dead oak branches included all standing and lying dead wood with a diameter greater than 7 cm and with a hard consistency based on resistance to finger pressure.

Canopy closure: Canopy closure at each study area was assessed by photographing the sky from the ground straight up. The sky was photographed on ten places with distances of 20 m. The photographed surface was ca $200\ m^2$. The photographs were analysed for the percentages of white (sky) and black (canopy) using ImageJ software (v.1.47). The percentage of the area of the sky that was black in the photographs was considered equivalent to the percentage of canopy closure.

Distance from stand boundary: The distance of each trap from the nearest boundary of the studied oak dominated forest stand (not the forest edge, stand means a homogeneous unit of the spatial distribution of the forest) was measured to the nearest meter.

Values of all variables are presented in Table 1.

### 2.4. Statistical Analysis

The importance of forest stand characteristics for ambrosia beetle abundance and species richness was evaluated by implementing a random forest algorithm using conditional inference trees as base learners provided in the party package (function cforest, 10,000 trees generated) in R 4.0.2 software (The R Foundation, Vienna, Austria). This method was used because it is a highly effective for evaluating the importance of explanatory variables, it can manage different types of variables, and is robust with respect to the multicollinearity of variables [38]. Because certain tested explanatory variables exhibited multicollinearity in our dataset, a conditional computation of the importance was performed (option conditional = TRUE). In addition, an unbiased random forest model was constructed (option control = cforest_unbiased), because the tested predictors were both quantitative and categorical variables. The response variable in the models was represented by the number of ambrosia beetle individuals and species in specific samples (sample =

beetles captured in one trap during per 2-week-period deployment as described earlier in the Methods). The statistical significance ($\alpha = 0.05$) of each explanatory variable was evaluated using the permutation-based attribute selection algorithm provided in the Boruta package. Finally, the marginal effect of the selected significant variables on the number of trapped ambrosia beetle individuals and species was visualised with a partial dependence plot using the package pdp (functions partial followed by plotPartial).

Ordination analyses of the relationship between ambrosia beetle abundance (i.e., species composition of their assemblages) and forest stand characteristics were performed in Canoco 5 (Wageningen University & Research, Wageningen, Netherlands). Based on preliminary analysis of the data (gradient length of response data was 2.2 SD units), redundancy analysis (RDA) was used. Data on the abundance of species were log-transformed and centered by species. After a global Monte-Carlo permutation test (10,000 permutations) of a full model (including all of the available explanatory variables) confirmed the overall significance of the relationship between response and explanatory variables (pseudo-$F$ = 4.2, $p$ = 0.001), a forward selection of explanatory variables was performed to identify the forest stand characteristics most closely associated with ambrosia beetle abundance and species richness.

## 3. Results

A total of 4179 individuals and 15 species of scolytid beetles were captured in the traps that had been deployed in 10 localities with oak trees in the western Czech Republic. Among the 15 species, six (represented by 4137 individuals) were ambrosia beetle species related to oak (Appendix A, Figure 3). *Anisandrus dispar* was the most abundant beetle trapped with an average of 40.0 individuals per sample. The 3520 specimens of *A. dispar* represented 84% of the trapped beetles (Appendix B). *Xyleborinus saxesenii* Ratzeburg, 1837 was the second most abundant species with an average of 6.5 individuals per sample. The 576 specimens of *X. saxesenii* represented 14% of the trapped beetles. The other four species were represented by fewer than 20 trapped specimens, i.e., they represented less than 1% of all trapped specimens (Appendix A, Figure 3).

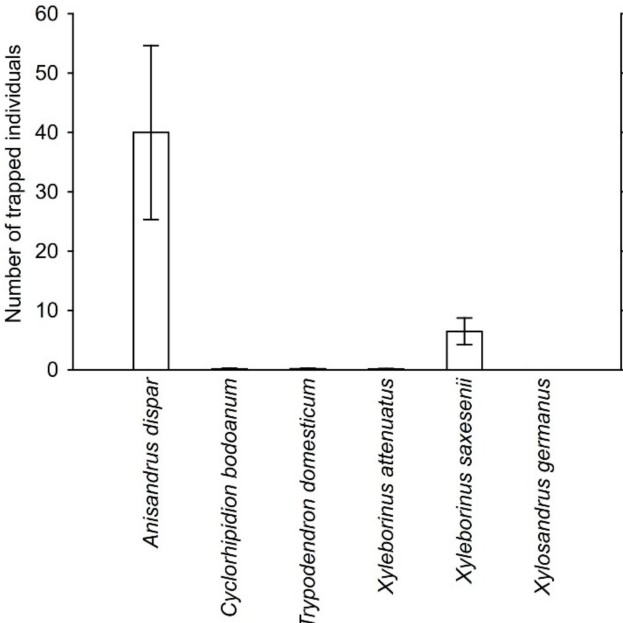

**Figure 3.** Number (mean $\pm$ SE) of individuals of ambrosia beetle species captured per sample. Each sample represented the beetles captured in one trap during 2-week period.

The number of ambrosia beetle individuals caught in the traps was significantly related to date of sampling, the percentage of canopy closure, and the abundance of dead oak branches per tree (Figure 4a). The number of ambrosia beetles trapped increased with

the percentage of canopy closure, but the increase was considerable only when canopy closure exceeded 45% (Figure 4b). The number of beetles caught also increased with the number of dead oak branches per tree, but the relationship plateaued with four dead oak branches per tree (Figure 4c).

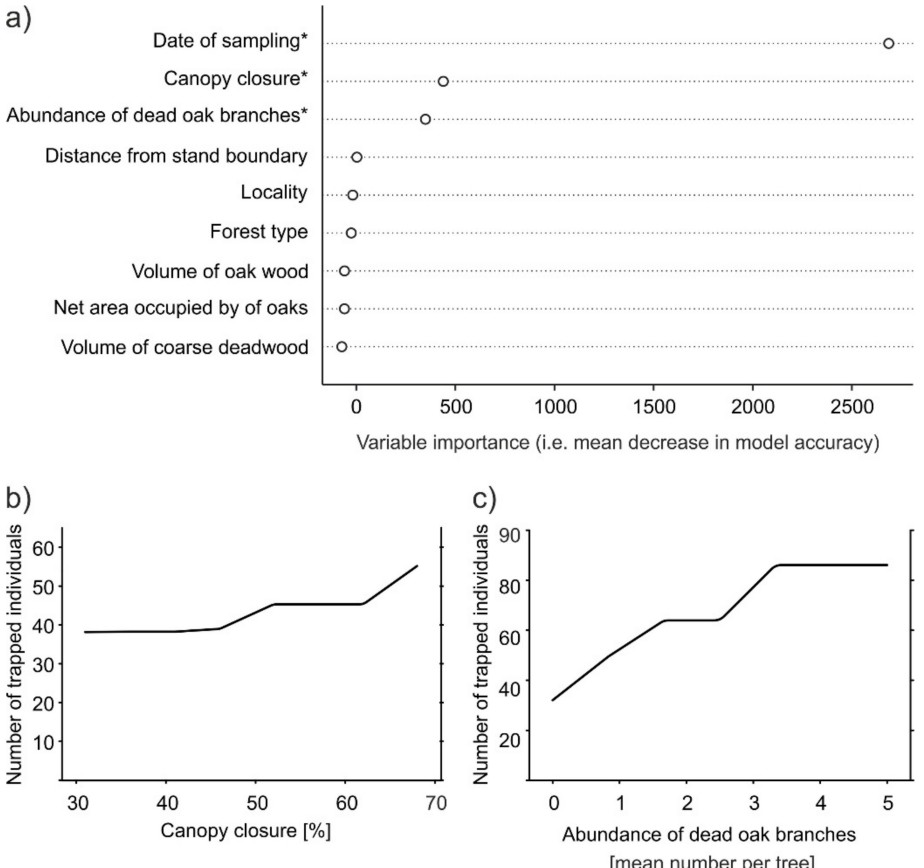

**Figure 4.** Results of random forest regression of number of ambrosia beetle individuals caught in traps as the response variable: (**a**) variable importance plot (based on the decrease of mean model accuracy with omission of the variable) showing the importance of particular variables for the number of ambrosia beetle individuals caught in the traps. Variables with a significant effect ($p < 0.05$) are denoted with an asterisk (*); (**b**,**c**) partial dependence plots showing the marginal effect of selected significant explanatory variables on the mean number of ambrosia beetle individuals trapped during the eight 2-week periods from April to August.

The number of ambrosia beetle species trapped was most strongly associated with the date of sampling (see Appendix B) and the percentage of canopy closure. Volume of oak wood and abundancy of dead oak branches were also associated with the number of ambrosia beetle species, whereas the associations with study plot and net area occupied by oaks were weak (even if statistically significant) (Figure 5a). The numbers of ambrosia beetle species trapped increased slightly with percentage of canopy closure, volume of oak wood, abundance of dead oak branches, and net area occupied by oaks (Figure 5b–e). The resulting curves were more or less sigmoidal, with most of the increase in the number of trapped species restricted to a narrow interval of explanatory variable values. This interval was between 40% and 50% for canopy closure; 200 and 250 m$^3$ for volume of oak wood; 2–3 for number of dead oak branches per tree; and 1.0–2.5 ha for the net area occupied by oaks.

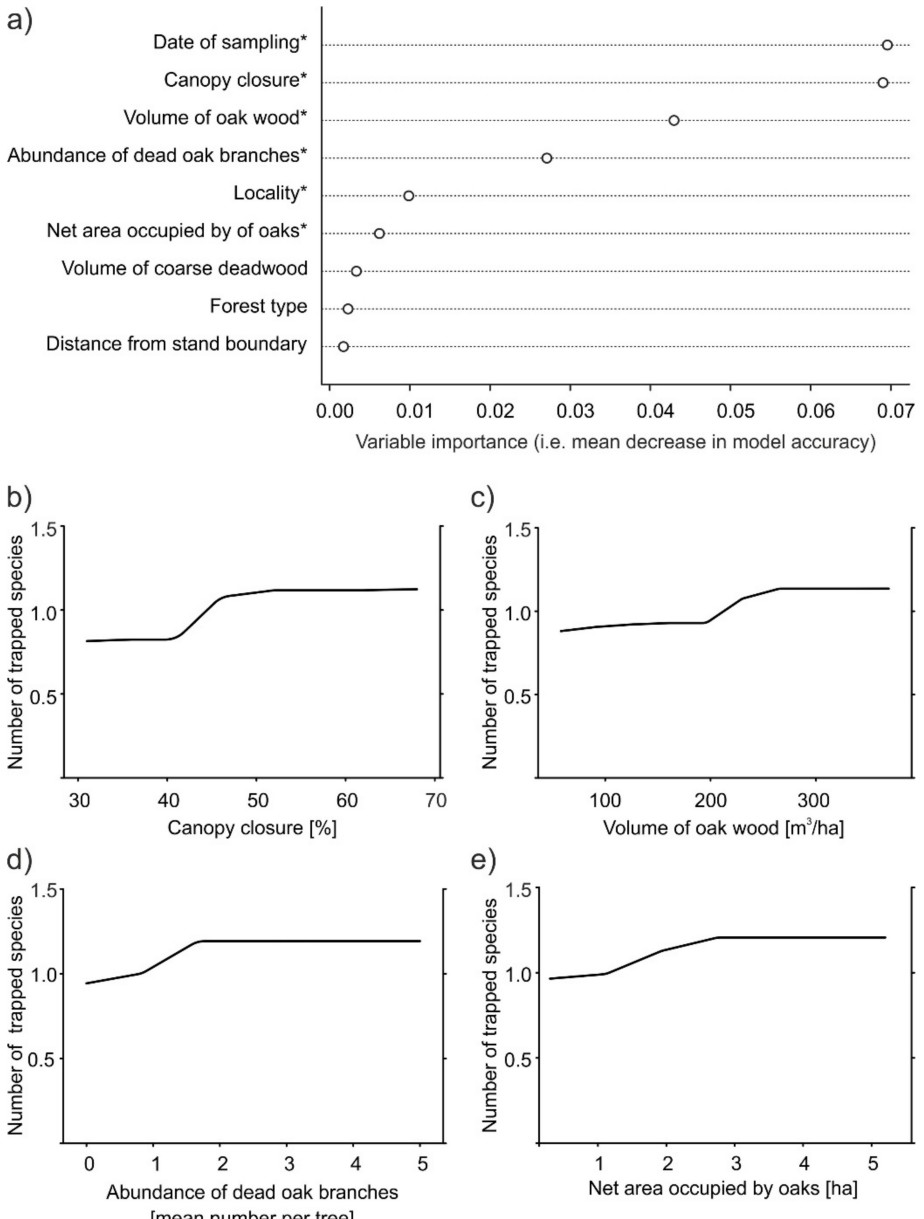

**Figure 5.** Results of random forest regression of the mean number of ambrosia beetle species captured per trap per 2-week period from April to August as the response variable: (**a**) plot of variable importance (based on the decrease of mean model accuracy with omission of the variable). Variables with a significant effect ($p < 0.05$) are denoted with asterisk (*); (**b**–**e**) partial dependence plots showing the marginal effect of the indicated explanatory variables on the mean number of ambrosia beetle individuals trapped during the eight 2-week periods from April to August.

The partial RDA analysis with the date of sampling treated as a covariable confirmed the significant associations between the studied explanatory variables and the occurrence and the numbers of ambrosia beetle individuals and species. Forward selection of explanatory variables indicated that two explanatory variables had significant effects: canopy closure and net area occupied by oaks (Table 2). Canopy closure had by far the highest explanatory power followed by net area occupied by oaks and the abundance of dead oak branches. The model including these three explanatory variables explained 24.4% of the variability in species occurrence and the numbers of ambrosia beetle individuals and species (Table 2).

**Table 2.** Results of the partial redundancy analysis forward selection of the percentage of variability in ambrosia species occurrence explained by the indicated forest stand variables. Conditional effects of the explanatory variables are shown. Effect of the date of trap deployment was removed by considering it as a covariable. The upper three explanatory variables highlighted in bold were included in the model based on the results of the forward selection process. Although it was not statistically significant, the abundance of dead oak branches was included in the model because it helped explain the variability in the occurrence of the species.

| Forest Stand Variable | Explained Variability in Species Occurrence (%) | Pseudo-*F* | *p* |
|---|---|---|---|
| **Canopy closure** | **16.5** | **15.6** | **0.002** |
| **Net are occupied by oaks** | **6.6** | **6.7** | **0.002** |
| **Abundance of dead oak branches** | **1.3** | **1.3** | **0.234** |
| Volume of coarse deadwood | 1.0 | 1.0 | 0.360 |
| Distance from forest stand boundary | 0.8 | 1.0 | 0.368 |
| Forest type | 0.4 | 0.5 | 0.652 |
| Volume of oak wood | 0.2 | 0.2 | 0.834 |

The RDA analysis revealed several associations between the abundance of ambrosia beetles and measured variables. The abundance of *A. dispar* was positively correlated with canopy closure and the abundance of dead oak branches. The abundances of *X. saxesenii* and the invasive species *X. attenuatus* and *C. bodoanum* were positively correlated with the net area occupied by oaks. The abundances of *T. domesticum* and *X. germanus* were positively but weakly correlated with the abundance of dead oak branches (Figure 6).

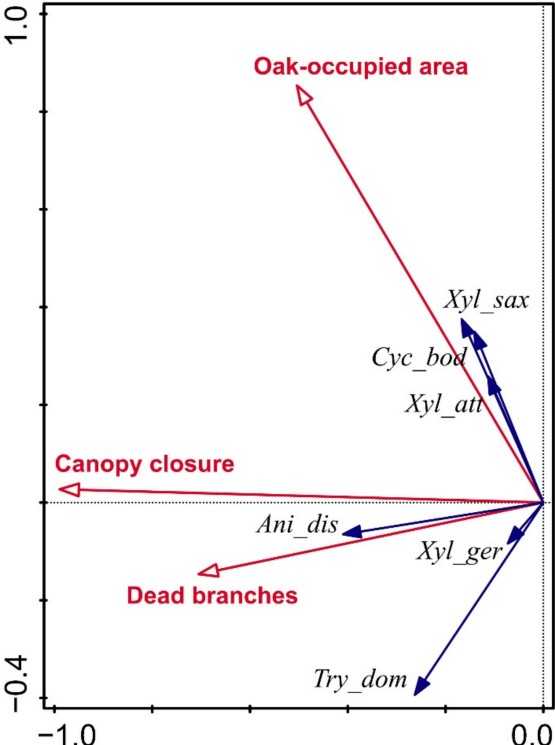

**Figure 6.** Results of redundancy analysis (RDA) of the relationship between presence of the ambrosia beetle species (blue arrows) and the most important characteristics of forest stand (red arrows). The species and the forest stand characteristics have been abbreviated to simplify the plot. The shown projection of the 1st and the 2nd axes represents 23.8% of the variability in species occurrence. Effect of the date of trap deployment was removed by considering it a covariable (*Ani_dis—Anisandrus dispar*, *Cyc_bod—Cyclorhipidion bodoanum*, *Try_dom—Trypodendron domesticum*, *Xyl_att—Xyleborinus attenuatus*, *Xyl_ger—Xylosandrus germanus*, *Xyl_sax—Xyleborinus saxesenii*).

## 4. Discussion

*Anisandrus dispar* and *X. saxesenii* were the most abundant ambrosia beetles detected in this study in western Bohemia (the Czech Republic) and are also very abundant in many other countries [9,39–42]. *Anisandrus dispar* is recognized as a serious pest of fruit and hazelnut trees [43]. It tends to infest trees that have been weakened by biotic and/or abiotic factors [44]. *X. saxesenii* is rarely considered to be a pest [43].

*Anisandrus dispar* is probably considered to be a pest more often than *X. saxesenii* because it can develop on thinner branches [44]. It is therefore able to attack hazelnut and fruit trees [43,44], which have thinner branches than forest trees. Given the high abundance of *A. dispar* in the current study, one would suppose that this species could cause substantial damage in the region, but such damage occurs only rarely and only on trees with thin stems [45]. Even in regions where *A. dispar* damages fruit trees, the damage it causes to oaks and beeches is insignificant [46], although it has been associated with oak damage [47].

Unlike *A. dispar* females, *X. saxesenii* females directly bore into tree trunks and form a radial entry tunnel [48]. *Xyleborinus saxesenii* females therefore require relatively thick branches and find relatively few resources in forests.

*Trypodendron domesticum* was the third most abundant species, but it was much less abundant than *A. dispar* and *X. saxesenii*. *Trypodendron* spp. attack the surfaces of tree trunks and thicker branches [37]. A high availability of suitable breeding substrate (e.g., wind-damaged or highly stressed trees) at the forest-stand scale seems to enhance *T. domesticum* population densities and attack rates, e.g., [49] break of bark beetles on spruce has resulted in the increased harvesting of spruce and the suspension of deciduous forest harvesting. As a result, suitable host trees are scarce in oak forests, and the abundance of *T. domesticum* has been low see also [50].

We found only a few individuals of the three species of invasive ambrosia beetles, i.e., *C. bodoanum*, *X. germanus*, and *X. attenuatus*. *Cyclorhipidion bodoanum* and *X. germanus* have recently spread from the west into the Czech Republic, and their abundance remains low [51,52]. In places where both *X. germanus* and *C. bodoanum* have established, however, they are the most abundant Scolytinae species [9,39,40,53].

The introduced ambrosia beetles are considered pests in Europe [54,55]. They can detect stress-induced ethanol emissions from weakened oak trees and can rapidly colonize those trees [56]. Once a forest begins to decline, trees lose vigor, which increases their susceptibility to secondary pests and pathogens [57–61]. If the abundances of invasive species increase overtime, which is likely, these invasive species are likely to contribute to the mortality of trees in weakened oak forests.

The number of ambrosia beetle individuals as well as ambrosia beetle species trapped in the traps was significantly related to date of sampling because of phenology beetles in our study (Figures 4 and 5). The time distribution of the two most abundant species (Appendix B) is in accordance with known seasonal flight activity of these two species [62–64]. The lower numbers of specimens caught in the second half of May are related to the rainy weather.

In this study, we found that the abundance of ambrosia beetles was significantly associated with the percentage of canopy closure and the abundance of dead oak branches (Figure 4). Rather than reflecting the preferences of all species of ambrosia beetles, these associations might mainly reflect the preferences of *A. dispar*, the dominant species in our study (Figure 3). On the other hand, increases in these two variables also increased the number of ambrosia beetle species trapped (Figure 5), suggesting that the percentage of canopy closure and the abundance of dead oak branches may affect species in addition to *A. dispar*. In support of that possibility, the abundance of *T. domesticum* was also positively correlated with the abundance of dead oak branches (there is a positive relationship also in *X. germanus*, but we have trapped only one specimen) (Figure 6). A high percentage of canopy closure is an indication of a stable and humid environment that is suitable for the growth of ambrosia fungi [65,66]. Because ambrosia bark beetles require these fungi as a

food source for development, both the fungi and the beetles are more frequent in wetter and warmer localities than in drier and colder localities [26–28]. In addition, ambrosia beetles generally prefer to inhabit the lower parts of tree canopies, such that most bark beetles are caught in traps at a height of 35–200 cm [36,67–71].

The volume of oak wood and the net area occupied by oaks represent the quantity and distribution of ambrosia beetle hosts. Ambrosia and other bark beetles are dependent on ephemeral and generally scattered breeding substrates [72,73], and it is therefore reasonable that the abundance of ambrosia beetle individuals and species at a site will increase with the concentration of host trees (Figures 5 and 6). The abundance of scolytids increases with resource availability [9,74]. In addition, the abundances of the invasive ambrosia beetles *X. attenuatus* and *C. bodoanum* were previously found to be positively correlated with the net area occupied by oaks, because both of these species live in oaks and other deciduous trees [40,49,50,75].

The number of species of ambrosia beetles trapped was significantly affected by study plot (Appendix A). On the other hand, the distance from the stand boundary had no effect on either the species spectrum or the number of captured beetles, which shows that a single trap was sufficient for monitoring the abundance of ambrosia beetle species and individuals at a study plot. This is reasonable because the beetles are lured to the traps by the bait, which was ethanol in the current study. Although not well studied, the population dynamics of ambrosia beetles are probably similar to those of other scolytids. Most individuals that hatched at a given site will probably tend to develop at the same site, but some individuals will disperse to search for new sites with suitable resources [73,76,77].

Although the number of ambrosia individuals trapped was not significantly associated with oak forest type (managed vs. unmanaged; Figure 4a), the abundance of many species was higher in the unmanaged forests than in the managed forests (Appendix A). We therefore cannot draw clear conclusions from these results. The unmanaged forests in the current study were abandoned coppicing forests in three cases (Vladař, Korunní, and Loket), a remnant of an old-growth forest that had been converted into a park in one case (Mariánské Lázně), and a reserve that resembled a virgin forest in only one case (Korunní) (Figure 1). We also captured many beetles at one managed site (Holubín) (Appendix A), which may help explain why the number of beetles captured was not significantly lower in the managed than in the unmanaged sites. Undisturbed, old-growth primary forests are generally considered to support high species richness [78], but species richness for some arthropod assemblages did not differ between primary forests and secondary or degraded (logged) forests in earlier studies, e.g., [79–82]. In a recent study, anthribid species richness did not significantly differ between primary and secondary forests [83], and anthribid species richness was greatly affected by the presence of suitable dead or dying fungus-infested wood, e.g., [84–86].

## 5. Conclusions

Six species of ambrosia beetles were recorded during the present study. The two most abundant species, *A. dispar* and *X. saxesenii*, represented 98% of the trapped beetles. Both of these ambrosia beetle species were more abundant in oak dominated forests with a high percentage of canopy closure, indicative of a stable and humid environment suitable for the growth of ambrosia fungi, compared to oak forests with a low percentage of canopy closure. Further, a higher abundance of dead oak branches in the canopy was found to be an important factor promoting the occurrence of *A. dispar*. Although the abundance of some species was slightly higher in unmanaged forests, no statistically significant differences in ambrosia beetle abundance in managed vs. unmanaged forests was found. We also have no evidence that particularly high abundances of *A. dispar* in the several studied unmanaged forests would lead to substantial damage to the surrounding forests. In the study area, the abundance of the recorded invasive ambrosia beetles (*C. bodoanum*, *X. attenuated* and *X. germanus*) was low but will probably increase over time. Once the availability of weakened trees increases locally, invasive ambrosia beetles could contribute to oak decline.

**Author Contributions:** Data curation, T.F.; formal analysis, J.H. and J.F.; methodology, J.H. and T.F.; writing—original draft, J.H., T.F. and J.F.; writing—review and editing, J.H., T.F. and J.F. All authors have read and agreed to the published version of the manuscript.

**Funding:** This research was supported by the grant "Advanced research supporting the forestry and wood-processing sector's adaptation to global change and the 4th industrial revolution", No. CZ.02.1.01/0.0/0.0/16_019/0000803 financed by OP RDE.

**Data Availability Statement:** The data presented in this study are available in Table 1 and Appendix A.

**Acknowledgments:** The authors thank Bruce Jaffee (USA) for editorial and linguistic improvement of manuscript; Denis Žižka, Zdeněk Fiala, and Miloš Fiala for support with field work; and Miloš Knížek (Praha) for confirmation of *C. bodoanum* identification.

**Conflicts of Interest:** The authors declare no conflict of interest.

## Appendix A

**Table A1.** Total numbers of bark beetles that were trapped at the 10 studied localities in the western Czech Republic (ambrosia beetle species that use oak trees as hosts are in bold).

| | Study Plot | | | | | | | | | | |
| Species | Vladař | Mariánské Lázně | Soos | Okounov | Korunní | Klára | Holubín | Loket | Holina | Lázně Kynžvart | Total |
|---|---|---|---|---|---|---|---|---|---|---|---|
| *Anisandrus dispar* Fabricius, 1792 | **122** | **70** | **2** | **1** | **296** | **2** | **226** | **810** | **1558** | **433** | **3520** |
| *Xyleborinus saxesenii* Ratzeburg, 1837 | **154** | **4** | **49** | | **15** | **5** | **57** | **194** | **3** | **95** | **576** |
| *Trypodendron domesticum* Linnaeus, 1758 | | **1** | | | | | | | **15** | | **16** |
| *Cyclorhipidion bodoanum* Reitter, 1913 | **13** | | | | | | | | | | **13** |
| *Trypodendron lineatum* Olivier, 1795 | | 8 | | | | | | | 1 | 2 | 11 |
| *Dryocoetes autographus* Ratzeburg, 1837 | | 5 | | | | | 1 | | 5 | | 11 |
| *Xyleborinus attenuatus* Blandford, 1894 | **9** | | | | | | | **1** | | | **10** |
| *Hylurgops palliatus* Gyllenhal, 1813 | | 6 | | | | | | | | | 6 |
| *Hylesinus varius* Fabricius, 1775 | | 4 | | | | | | | | 1 | 5 |
| *Pityogenes chalcographus* Linnaeus, 1761 | | 1 | | | 1 | | | | | 1 | 3 |
| *Polygraphus grandiclava* C.G. Thomson, 1886 | | | | | 1 | 1 | | | | | 2 |
| *Polygraphus poligraphus* Linnaeus, 1758 | 1 | | | | | | 1 | | | | 2 |
| *Hylastes cunicularius* Erichson, 1836 | | | | | | | | | 1 | | 1 |
| *Scolytus intricatus* Ratzeburg, 1837 | | | | | | | | 1 | | | 1 |
| *Xylosandrus germanus* Blandford, 1894 | | | | | | | | | | **1** | **1** |
| *Ips typographus* Linnaeus, 1758 | | 1 | | | | | | | | | 1 |
| Total | 299 | 100 | 51 | 1 | 313 | 7 | 286 | 1006 | 1583 | 533 | 4179 |

**Appendix B**

**Figure A1.** Number (mean ± SE) of individuals of the two most abundant ambrosia beetle species captured per sample in particular sampling periods from April to August 2020. Each sample represented the beetles captured in one trap during 2-week period.

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
