# Peer review of "Ambrosia Beetles Prefer Closed Canopies: A Case Study in Oak Forests in Central Europe"

_forests, doi:10.3390/f12091223_

Round 1

Reviewer 1 Report

The topic proposed in the manuscript is interesting because the authors try to find a link between the dynamic population parameters of ambrosia beetles and oak stands canopy closure.
Unfortunately, in the Introduction, the authors do not provide enough information for the reader to form a clear idea about the issue addressed. The style is narrative, without emphasizing a particular problem or solutions identified by other authors, despite the fact that the objectives of the study are clearly and concisely defined.
And as for the description of the methods and materials used, they are obviously treated briefly, giving the manuscript a superficial look. For example, the description of study areas is incomplete and partially ambiguous. This lacks important information on key factors such as thermal and rainfall regime (e.g. the average amount of annual precipitation in each experimental surface, temperature, etc.); orographic condition: altitude but especially exposure; but also the intrinsic parameters of the studied stands such as the composition of the stands, age, and dendrometric features.
In addition, I find it inappropriate to use the term locality to the detriment of experimental / studied / plot areas.
The manufactured lures seem to be an artisanal way to trap insects. If authors can not reference the method then should clearly explain why they use this and for what species is appropriate; Authors should specify the range of action (effective trapping) of such assemblage and how often did they replace lures. I don’t see any control in the field experiments.
When dividing the stands into managed and unmanaged, the authors must specify, in the case of the former, whether at the time of the experiments there were tree remains from recent forestry or logging operations and the year when the last operation was carried out.
Is it not very clear how the canopy closure was evaluated, how big were the photographed surfaces (projection on the ground)? Or how many photos were taken on each surface? How were places where the photos were taken chosen?
I am afraid that I don't understand what the authors are trying to say related to “Distance from stand boundary” (line 162-164).
Part of the result seems honest and fair presented, but some, like for example distribution of the captures over time need to be detailed in order to understand how these evolved during the season. If possible, I suggest authors correlate the captures with precipitations and temperatures or slope exposure and stand composition.
Related to the Discussion, I believe that the authors should revise the sentence from lines 315-316 - this supposition can't be supported by only one insect record (e.g 1-specimen of X.germanus traped).
I consider that the manuscript needs a full English and style revision (especially for correcting redundancy in the sentences). Also, some writing errors were identified (line 12-13; line 228-229, line 305).

Reviewer 2 Report

  1. Bring the Abstract section into compliance with the journal rules.
  2. Line 64-67. You can add references: Kovach J. & Gorsuch C.S., 1985, Survey of ambrosia beetle species infesting South Carolina peach orchards and a taxonomic key for the most common species. Journal of Agricultural Entomology 2: 238–247.; https://dx.doi.org/10.24189/ncr.2020.020; Levesque C. & Levesque G.Y., 2019, A five-year study of the flying beetles (Coleoptera) from a grassland and an adjacent woods in Southern Québec (Canada). Great Lakes Entomolo-gist 52 (1-2): 45-52.; Leksono, A.S.; Takada, K.; Koji, S.; Nakagoshi, N.; Anggaeni, T.; Nakamura, K. Vertical and seasonal distribution of flying beetles in a suburban temperate deciduous forest collected by water pan trap. Insect Sci. 2005, 12, 199-206.
  3. Line 115. Specify the number of traps that stood in each area.
  4. Make a more detailed description of "Environmental variables".
  5. Line 272-276. You need to add some references: https://dx.doi.org/10.24189/ncr.2020.008; https://doi.org/10.3390/insects12050407; Skvarla, M.J.; Dowling, A.P.G. A comparison of trapping techniques (Coleoptera: Carabidae, Buprestidae, Cerambycidae, and Curculionoidea excluding Scolytinae). J. Insect Sci. 2017, 17, 1–28.; Saruhan I. & Akyol H., 2012, Monitoring population density of Anisandrus dispar and Xylebor-inus saxesenii (Coleoptera: Scolytinae, Curculionidae) in hazelnut orchards. African J. Biotechnol. 11: 4202–4207.; doi: 10.18698/2542-1468-2018-5-34-41.
  6. Line 297-299. How can you explain the low number of some species of ambrosia beetles? Is there no influence of baits in traps here? Maybe because of the bait, some species fly better, and other species are lured worse?
  7. Line 337. The authors do not know the publication - http://dx.doi.org/10.12775/EQ.2021.004. This publication describes the dynamics of the number of one species of ambrosia beetle and there are references on this topic.
  8. The conclusion needs to be redone. Enter all the information that the authors received in the study. For example, the number of species and other specific research results.

Round 2

Reviewer 2 Report

I thank the authors for the corrections.